# Predictors of Hypoxemia and Related Adverse Outcomes in Patients Hospitalized with COVID-19: A Double-Center Retrospective Study

**DOI:** 10.3390/jcm10163581

**Published:** 2021-08-14

**Authors:** Rabea Asleh, Elad Asher, Oren Yagel, Tal Samuel, Gabby Elbaz-Greener, Arik Wolak, Ronen Durst, Eli Ben-Chetrit, Ran Nir-Paz, Yigal Helviz, Limor Rubin, Ariella Tvito, Michael Glikson, Offer Amir

**Affiliations:** 1Heart Institute, Hadassah Medical Center, Faculty of Medicine, Hebrew University of Jerusalem, Jerusalem 91120, Israel; oreny@hadassah.org.il (O.Y.); gabby@hadassah.org.il (G.E.-G.); durst@hadassah.org.il (R.D.); oamir@hadassah.org.il (O.A.); 2The Jesselson Integrated Heart Center, Shaare Zedek Medical Center, Hebrew University of Jerusalem, Jerusalem 91031, Israel; easher@szmc.org.il (E.A.); talsamuel81@gmail.com (T.S.); arikwt@szmc.org.il (A.W.); elibc@szmc.org.il (E.B.-C.); yigalhe@szmc.org.il (Y.H.); ariellat@szmc.org.il (A.T.); mglikson@szmc.org.il (M.G.); 3Department of Clinical Microbiology and Infectious Disease, Hadassah University Medical Center, Faculty of Medicine, Hebrew University of Jerusalem, Jerusalem 91120, Israel; nirpaz@hadassah.org.il; 4Department of Allergy and Clinical Immunology, Hadassah University Medical Center, Faculty of Medicine, Hebrew University of Jerusalem, Jerusalem 91120, Israel; limorlaz@hadassah.org.il; 5Department of Medicine, Azrieli Faculty of Medicine, Bar-Ilan University, Safed 13403, Israel

**Keywords:** COVID-19, blood oxygen saturation, hypoxemia, body mass index, body surface area, obesity, predictors, outcome

## Abstract

Hypoxemia is a hallmark of coronavirus disease 2019 (COVID-19) severity. We sought to determine predictors of hypoxemia and related adverse outcomes among patients hospitalized with COVID-19 in the two largest hospitals in Jerusalem, Israel, from 9 March through 16 July 2020. Patients were categorized as those who developed reduced (<94%) vs. preserved (≥94%) arterial oxygen saturation (SpO_2_) within the first 48 h after arrival to the emergency department. Overall, 492 hospitalized patients with COVID-19 were retrospectively analyzed. Patients with reduced SpO_2_ were significantly older, had more comorbidities, higher body surface area (BSA) and body mass index (BMI), lower lymphocyte counts, impaired renal function, and elevated liver enzymes, c-reactive protein (CRP), and D-dimer levels as compared to those with preserved SpO_2_. In the multivariable regression analysis, older age (odds ratio (OR) 1.02 per year, *p* < 0.001), higher BSA (OR 1.16 per 0.10 m^2^, *p* = 0.003) or BMI (OR 1.05 per 1 kg/m^2^, *p* = 0.011), lower lymphocyte counts (OR 1.72 per 1 × 10^3^/μL decrease, *p* = 0.002), and elevated CRP (1.11 per 1 mg/dL increase, *p* < 0.001) were found to be independent predictors of low SpO_2_. Severe hypoxemia requiring ventilatory support, older age, and pre-existing comorbidities, including underlying renal dysfunction and heart failure, were found to be significantly associated with in-hospital mortality. These findings suggest that assessment of predictors of hypoxemia early at the time of hospitalization with COVID-19 may be helpful in risk stratification and management.

## 1. Introduction

Coronavirus disease 2019 (COVID-19), caused by severe acute respiratory syndrome coronavirus 2 (SARS-CoV-2), has become a global pandemic with enormous morbidity and mortality burdens. Risk factors for severe outcomes from COVID-19 infection may include older age, male gender, and chronic underlying health conditions, such as diabetes mellitus (DM), hypertension, obesity, ischemic heart disease, and heart failure (HF) [1,2,3,4,5]. Although substantial research has been undertaken to identify important risk factors of worse outcomes associated with the COVID-19 pandemic, there is still a need for better understanding of important predictors of in-hospital clinical deterioration early at the time of diagnosis of the disease. Given the limited availability of critical care resources in many countries that are still struggling to contain the disease, improvement in risk stratification of COVID-19 patients is clinically imperative for patients’ triage and optimal management.

Development of respiratory complications during hospitalization with COVID-19, ranging from mild hypoxemia to acute respiratory distress syndrome (ARDS) requiring intubation and mechanical ventilation, is an important clinical hallmark of the disease severity [6]. Noninvasive pulse oxygen saturation (SpO_2_) is an important marker of disease severity in accordance with World Health Organization (WHO) guidelines [7]. Specifically, SpO_2_ < 94% has been associated with moderate to severe disease and high morbidity as compared to patients with preserved blood oxygen saturation (SpO_2_ ≥ 94%) [8,9]. In moderately to critically ill patients with COVID-19-associated pneumonia, dyspnea and severe hypoxemia (SpO_2_ < 90% despite oxygen supplementation) were associated with increased in-hospital mortality [10]. Moreover, many COVID-19 patients who initially presented with hypoxemia without signs of respiratory distress (silent hypoxemia) may later develop respiratory failure requiring mechanical ventilatory support [6,11,12]. Although hypoxemia can serve as an important intermediate endpoint for poor outcomes in COVID-19 patients, important predictors of increased oxygen requirements that may be associated with adverse outcomes during hospitalization have not been adequately investigated. Early identification of simple but reliable predictors of hypoxemia among hospitalized COVID-19 patients may thus help in risk stratification and management before clinical deterioration occurs.

Therefore, in the present double-center study, we sought to investigate whether there might be unique predictors of the development of hypoxemia during hospitalization due to COVID-19 and to examine their associations with subsequent adverse clinical outcomes. 

## 2. Materials and Methods

### 2.1. Study Design and Patients

This double-center retrospective cohort study enrolled adult patients (age ≥18 years of both genders) with COVID-19 hospitalized at the only two tertiary medical centers in Jerusalem, Hadassah University Medical Center and Shaare Zedek Medical Center, from 9 March through 16 July 2020. The study was conducted according to the guidelines of the Declaration of Helsinki and approved by the Institutional Review Boards of the two medical centers, Hadassah University Medical Center (ethics committee number: HMO-0460-12) and Shaare Zedek Medical Center (ethics committee number: SZMC-0158-20). Data were collected separately at each institution and were later submitted into an electronic case report form (eCRF).

Patients were included in the study if they met the following criteria: confirmed COVID-19 infection based on positive nasopharyngeal swab real-time reverse transcriptase polymerase chain reaction (RT-PCR) testing; typical symptoms of a respiratory infection/pneumonia, i.e., fever, cough, dyspnea, etc.; and objective evidence of new-onset pulmonary infiltrates on chest computed tomography (CT) requiring hospitalization. For the purpose of our study, patients were categorized according to their SpO_2_ levels for assessment of disease severity within 48 h after arrival to the emergency department: mild/moderate (SpO_2_ ≥ 94%) or severe (SpO_2_ < 94%) disease as previously described [7,8] and as consistent with the WHO classification [9]. 

### 2.2. Clinical and Laboratory Data and Outcome Assessment

Demographic and clinical characteristics were collected from electronic medical records, medical history, and physical examination findings. Measures of complete blood count (CBC) and white blood cell differential, basic metabolic panel, and markers of inflammation, including CRP and D-dimer, were all acquired at the time of admission to the hospital. Weight and height were used to calculate the body surface area (BSA) measures according to the Mosteller formula [13]: BSA = 0.16667 * weight^0.5^ * height^0.5^, and body mass index (BMI) measures were calculated according to the formula: BMI = weight/height^2^. Estimated glomerular filtration rate (eGFR) was calculated using the Cockcroft–Gault (CG) formula. For analyzing the length of stay in hospital and in-hospital mortality, follow-up duration was recorded as the interval (in days) from the date of admission to the date of discharge or the date of death, whichever occurred first. Acute respiratory distress syndrome (ARDS) was defined according to the Berlin Criteria as described previously [14].

### 2.3. Statistical Analysis

Data were checked for accuracy and out-of-range values by the coordinating unit prior to statistical analysis. Categorical variables were expressed as frequency and percentage, and continuous variables were expressed as mean ± standard deviation (SD) or as median and interquartile range (IQR) where appropriate. Patient characteristics were compared between those with and without reduced oxygen saturation (SpO_2_ < 94%) using the chi-square test for categorical variables (or Fischer’s exact test if the expected count was below 5), t test or analysis of or variance (ANOVA) for normally distributed continuous variables, and Wilcoxon signed rank or Kruskal–Wallis tests for continuous variables with skewed distribution. Univariate followed by multivariate logistic regression models were constructed to identify factors associated with hypoxia. In the multivariable model, we used a backward stepwise analysis by including all variables with *p*-values < 0.1 obtained in the univariate model. Results were expressed as odds ratios (ORs) with 95% confidence intervals (CIs). We also used a receiver operating characteristic (ROC) curve analysis to determine the optimal cutoff values of the independent variables for predicting low SpO_2_ based on the threshold yielding the best combination of sensitivity and specificity. A Cox regression model was run to determine associations with in-hospital mortality, expressed as hazard ratios (HRs) with 95% CIs. Data were analyzed using the JMP software, Version 14.1 (SAS Institute, Inc, Cary, NC, USA), with all statistical tests conducted at the 5% significance level.

## 3. Results

### 3.1. Patient Characteristics

Overall, 683 COVID-19 confirmed patients were admitted to either one of the two major hospitals in Jerusalem during the study period. After screening, 492 patients were found to have available data at baseline and thus included in the study analysis. When stratified by arterial oxygen saturation, 221 (45%) were found to develop hypoxemia (SpO_2_ < 94%) and 271 (55%) had preserved oxygen saturation (SpO_2_ ≥ 94%) during hospitalization. Baseline demographic, clinical, and laboratory characteristics and treatment information of the overall patient cohort and a comparison of those with and without low SpO_2_ (<94%) are presented in Table 1. The mean (±SD) age was 55.9 ± 20.6 years, and 259 patients (52.6%) were males. The mean BSA and BMI was 1.91 ± 0.24 m^2^ and 28.20 ± 6.10 kg/m^2^, respectively. Regarding comorbidities, 42% of patients had a history of hypertension, 28% had hyperlipidemia, 29% had DM, 18.8% had a cognitive decline, and 17.8% had atherosclerotic cardiovascular disease (CVD).

Compared to patients with preserved oxygen saturation (SpO_2_ ≥ 94%), patients with low oxygen saturation (SpO_2_ < 94%) were older (62.3 ± 17.8 vs. 50.7 ± 21.2 years; *p* < 0.001), were more likely to be males (61% vs. 46%; *p* < 0.001), and had higher BSA (1.96 ± 0.23 vs. 1.87 ± 0.25; *p* < 0.001) and BMI (29.61 ± 5.92 vs. 27.13 ± 6.10 kg/m^2^; *p* < 0.001) (Table 1). Furthermore, patients with low SpO_2_ were more likely to have hypertension and DM compared with patients with preserved SpO_2_, while the distribution of the other comorbidities was not significantly different between the two groups (Table 1). Treatment with statins, ACE inhibitors/ARBs, antiplatelets, insulin, and diuretics at baseline was significantly more common in patients with low SpO_2_ (*p* < 0.05 for all), whereas the use of other medications was similarly distributed between the two groups. Analysis of laboratory data showed that patients with low SpO_2_ had decreased lymphocyte counts, but similar white blood cell counts, as compared to those with preserved SpO_2_. Patients with low SpO_2_ had significantly higher blood urea nitrogen (BUN), creatinine, aspartate transaminase (AST), gamma-glutamyl transferase (GGT), and D-dimer and lower eGFR than those without low SpO_2_. CRP levels were markedly increased in the low SpO_2_ group (7.3 (4.0–13.5) vs. 1.5 (0.4–5.2); *p* < 0.001) compared with the preserved SpO_2_ group (Table 1).

### 3.2. Predictors of Low Oxygen Saturation

Univariable and multivariable regression models were constructed to examine the association of baseline demographic and clinical parameters with development of low SpO_2_ during hospitalization with COVID-19 (Table 2). Older age (OR 1.03 per year; *p* < 0.001), male gender (OR 1.86; *p* < 0.001), higher BMI (OR 1.08 per 1 kg/m^2^; *p* < 0.001), higher BSA (OR 1.17 per 0.1 m^2^; *p* < 0.001), DM (OR 2.00; *p* = 0.002), and hypertension (OR 2.11; *p* < 0.001) were found to be significant predictors of low SpO_2_ in the univariable analysis. Furthermore, treatment with ACE inhibitors/ARBs (OR 1.75; *p* = 0.027) and diuretic therapy (OR 2.63; *p* = 0.004) were significantly associated with low SpO_2_. Among the laboratory parameters, blood urea nitrogen (BUN) (OR 1.03 per mg/dL; *p* = 0.001), CRP (OR 1.15 per mg/dL; *p* < 0.001), D-dimer (OR 1.04 per 0.10 ug/mL; *p* < 0.001), fibrinogen (OR 1.05 per 10 ng/mL; *p* < 0.001), AST (OR 1.12 per 10 IU/L; *p* = 0.004), and lower lymphocyte count (OR 2.56 per 10^3^/µL decrease; *p* < 0.001) at baseline were significant predictors of low SpO_2_.

In the multivariable regression model, which included all variables with *p* value < 0.1 obtained from the univariate model, older age (OR 1.02 per year, 95% CI: 1.01–1.03; *p* < 0.001), higher BMI (OR 1.05 per kg/m^2^, 95% CI: 1.01–1.09; *p* = 0.011), lower lymphocyte count (OR 1.72 per 10^3^/µL decrease, 95% CI: 0.41–0.81; *p* = 0.002), and elevated CRP (OR 1.11 per 1 mg/dL, 95% CI: 1.07–1.16; *p* < 0.001) were independent predictors of low SpO_2_ (Table 2). The association between gender and hypoxemia was attenuated after adjustment for other variables (OR 1.50, 95% CI: 0.96–2.32; *p* = 0.074, for males vs. females). When BSA, instead of BMI, was included in the multivariable regression model, the same independent predictors, including older age (OR 1.02 per year; *p* < 0.001), lower lymphocyte count (OR 0.58 per 10^3^/µL; *p* = 0.002), and elevated CRP (OR 1.11 per 1 mg/dL; *p* < 0.001) were obtained. Higher BSA was also found to be a significant predictor of low SpO_2_ (OR 1.16 per 0.10 m^2^, 95% CI: 1.05–1.27; *p* = 0.003) in this model (Table 2). The correlations of BMI vs. BSA with SpO_2_ as a continuous variable are provided in the Appendix A.

### 3.3. Receiver Operating Characteristic (ROC) Curves for the Diagnosis of low SpO_2_

ROC curve analysis for the independent predictors of low SpO_2_ found in the multivariate model showed that the individual area under the curve (AUC) for age, BMI, BSA, lymphocyte count, and CRP was 0.67, 0.65, 0.64, 0.71, and 0.79, respectively. The optimal cutoffs (with sensitivity and specificity) for predicting low SpO_2_ of these variables were as follows: age of 50 years (yielding a sensitivity of 75% and specificity of 50%), BMI of 27.4 kg/m^2^ (65% and 62%), BSA of 1.9 m^2^ (61% and 60%), lymphocyte count of 1400 cells/µL (77% and 59%), and CRP of 4 mg/dL (76% and 71%). When age, BMI, lymphocyte counts, and CRP were all included in the ROC analysis, they demonstrated improved diagnostic performance for low oxygen saturation (AUC 0.81 (95% CI: 0.77–0.85) with a sensitivity and specificity rate of 87% and 62%, respectively, *p* < 0.001). Substitution of BMI with BSA in multivariate ROC analysis resulted in similar diagnostic performance (AUC 0.80 (95% CI: 0.76–0.84) with a sensitivity and a specificity rate of 80% and 70%, respectively, *p* < 0.001). Using the optimal cutoffs of the variables found to be independently associated with low SpO_2_, we found the mean (SD) oxygen saturation was significantly lower among patients at age ≥ 50 (92.1% ± 4.6% vs. 94.5% ± 4.3%; *p* < 0.001), with BMI ≥ 27.4 kg/m^2^ (92.0% ± 5.3% vs. 94.1% ± 3.6%; *p* < 0.001) or BSA ≥ 1.9 m^2^ (92.1% ± 5.2% vs. 94.0% ± 3.8%; *p* < 0.001), lymphocyte count <1400 cells/μL (91.7% ± 5.2% vs. 94.6% ± 3.1%; *p* < 0.001), and those with CRP ≥ 4.0 mg/dL (90.9% ± 5.4% vs. 94.8% ± 2.9%; *p* < 0.001) (Figure 1).

Patients were categorized into five subgroups according to a score generated by calculating the number of risk factors (age ≥ 50, BMI ≥ 27.4, lymphocyte count <1400, and CRP ≥ 4) in each patient. As shown in Figure 2a, the higher the risk score calculated for each patient, the lower the SpO_2_ was observed. Furthermore, the proportion of patients with SpO_2_ < 94% increased according to the number of risk factors; 2.4% of patients who had none of these variables compared with 80.3% of those with all risk factors were found to develop low oxygen saturation (Figure 2b). Similar results were obtained when BSA was used instead of BMI in this score model (Appendix A).

### 3.4. Respiratory-Related Adverse Outcomes

Of the 221 patients developing early hypoxemia (SpO_2_ < 94%) due to COVID-19, 41 (18.6%) required ventilation (14 (6.3%) noninvasive and 27 (12.2%) invasive ventilation) and 22 (10.0%) developed ARDS during hospitalization. By comparison, ventilation was required in 11 (4.1%) patients (6 (2.2%) noninvasive and 5 (1.9%) invasive) and ARDS developed in 4 (1.5%) patients among those who did not have early low oxygen saturation. Consequently, early identification of SpO_2_ < 94% was associated with increased odds of ventilation (OR 5.5, 95% CI: 2.7–10.8; *p* < 0.001) and development of ARDS (OR 7.4, 95% CI: 2.5–21.8; *p* < 0.001) compared with preserved SpO_2_. Length of stay in hospital was significantly longer in patients with SpO_2_ <94% as compared to patients with SpO_2_ ≥ 94% (8 (5–13) vs. 4 (2–8) days; *p* < 0.001). 

Consistent with the predictors of low SpO_2_, patients requiring ventilation during hospitalization were more likely to be older (16.0% vs. 2.1% of patients were at age ≥ 50 years; *p* < 0.001), with higher BMI (13.8% vs. 7.3% with BMI >27.4 kg/m^2^; *p* = 0.019) and a trend towards higher BSA (13.0% vs. 8.1% with BSA > 1.9 m^2^; *p* = 0.079), lower lymphocyte count (14.2 vs. 6.0% with lymphocyte count <1400 cells/µL; *p* = 0.004), and low CRP levels (19.0% vs. 3.6% with CRP ≥ 4.0 mg/dL; *p* < 0.001) (Figure 3a). The probability of requiring ventilation was positively correlated with the number of risk factors consisting of the predictors above (0.0%, 2.2%, 5.1%, 18.2%, and 23.9% of patients requiring ventilation for having 0, 1, 2, 3, and 4 risk scores, respectively; P_trend_ < 0.001) (Figure 3b). Similarly, there was a higher probability of developing ARDS among those with higher number of risk factors (P_trend_ < 0.001), with approximately one out of six patients with all risk factors (age ≥ 50 years, BMI ≥ 27.4 kg/m^2^, lymphocyte count ≥1400 cells/µL, and CRP ≥ 4.0 mg/dL) developing ARDS vs. none of those without any or only one of these risk factors. Finally, LOS was also significantly longer in the presence of higher risk score (median: 3, 3, 5, 8, and 9 days for patients with 0, 1, 2, 3, and 4 risk factors, respectively; P_trend_ < 0.001) (Figure 3c).

### 3.5. Associations of Clinical Characteristics and Comorbidities with In-Hospital Mortality 

In total, 33 patients (6.7%) died during hospitalization for COVID-19. The median (IQR) LOS in hospital for the overall cohort was 6 (3–9) days (5 (3–10) days vs. 10 (6–16) days for those who survived and those who died during hospitalization, respectively; *p* < 0.001). In-hospital mortality occurred more frequently in patients with low oxygen saturation requiring ventilation (3.9%, 30.0%, and 31.3% mortality in those required none, noninvasive, or invasive ventilation, respectively; P_trend_ < 0.001) and those who developed ARDS (5.2% vs. 34.6%; OR 9.8, 95% CI: 3.9–24.1; *p* < 0.001). Invasive ventilation was associated with an 8.6-fold increase in the probability of in-hospital mortality compared with patients who did not require ventilation or were treated only with noninvasive ventilation (OR 8.6, 95% CI: 3.7–20.4; *p* < 0.001).

In the univariate Cox regression analysis, age 60 or older (hazard ration (HR) 15.7, 95% CI: 2.1–115.8; *p* < 0.001), female gender (HR 2.1, 95% CI: 1.1–4.3; *p* = 0.039), underlying HF (HR 4.1, 95% CI: 1.8–9.1; *p* < 0.001), established CVD (HR 2.3, 95% CI: 1.1–5.1; *p* = 0.036), and the presence of any comorbidity (defined as ≥1 of the following comorbidities: hypertension, DM, established CVD, underlying HF, or chronic renal failure stage ≥4) (HR 17.0, 95% CI: 2.3–126.7; *p* < 0.001), but not BMI (HR 1.4, 95% CI: 0.7–2.8; *p* = 0.392) or BSA (HR 0.62, 95% CI: 0.31–1.20; *p* = 0.175), were significantly associated with increased in-hospital mortality (Figure 4 and Table 3). A trend towards increased mortality among patients with hypertension was observed (HR 2.0, 95% CI: 0.9–4.6; *p* = 0.090). After adjustment for age and gender, heart failure (HR 2.7, 95% CI: 1.2–6.0; *p* = 0.016), chronic renal failure (HR 2.8, 95% CI: 1.3–6.0; *p* = 0.012), and the presence of any comorbidity (HR 9.4, 95% CI: 1.2–75.4; *p* = 0.035) remained significantly associated with an increased risk of in-hospital mortality, while the associations of hypertension and CVD with mortality were attenuated (Table 3). High BMI, BSA, or CRP were not associated with in-hospital mortality. However, hypoxemia requiring ventilation was found to be significantly associated with increased risk of mortality (adjusted HR 2.3, 95% CI: 1.1–4.9; *p* = 0.036) (Table 3).

## 4. Discussion

While most studies have focused on predictors of mortality among patients affected by COVID-19, our study provides insights into unique and clinically plausible risk factors for hypoxemia. By considering hypoxemia as an intermediate outcome, which could be initially silent until deterioration may occur, assessment of these risk factors may be helpful in more appropriate patient triage and aggressive treatment with oxygen supplementation to mitigate disease progression. In this current retrospective double-center study involving hospitalized patients with COVID-19, we report on several key observations regarding risk factors for hypoxemia and subsequent related outcomes in a relatively large sample of moderately to severely ill patients with COVID-19. Our study highlights several salient findings. First, the present data demonstrated that older age, obesity, lymphopenia, and elevated CRP were independent predictors of hypoxemia that occurred early after hospitalization with COVID-19. Besides the individual predictive value of each of these risk factors, the presence of multiple factors in the same patient had an improved diagnostic performance of hypoxemia in this population. Second, BSA was found to be a noninferior measure of body habitus in predicting hypoxemia when compared to BMI, and therefore, it may serve as an additional important prognostic indicator for COVID-19 severity. Third, hypoxemia was associated with respiratory-related adverse events, including requirement for ventilation and development of ARDS, and longer in-hospital LOS. Finally, we found that severe hypoxemia requiring ventilatory support as well as other comorbidity factors, including older age, chronic renal failure, and underlying HF, were significantly associated with increased in-hospital mortality.

COVID-19 pneumonia is mainly characterized by progressive hypoxemia that can ultimately lead to ARDS and high mortality rates. Among the potential mechanisms underlying the development of low oxygen saturation in COVID-19 patients are potentiated immune response to viral infiltration, inflammation and proinflammatory cytokine release, pulmonary vasoconstriction, and COVID-19-related intravascular thrombosis and pulmonary emboli [15,16]. On the other hand, there is a growing body of evidence that hypoxemia per se may be an important contributor to, rather than simply a marker for, progressive lung injury [10,16,17]. Hypoxemia itself may thus beget worsening hypoxemia by providing a positive feedback loop that potentiates lung injury and eventually leads to respiratory-related adverse events and death. In support of this notion, hypoxia has been implicated in promotion of viral replication [18,19], lung inflammation [16,17], pulmonary vasoconstriction [20], and intravascular thrombosis [21,22]. Taken together, hypoxemia may act as an amplifier of the COVID-19 disease process, which may plausibly be predicted and interrupted by earlier initiation of preventive and management measures. 

The association of inflammatory markers with high oxygen requirement has not been adequately studied. In 84 Japanese patients admitted with mild to moderate COVID-19, advanced age, lymphopenia, and obesity, but not CRP, were associated with increased oxygen requirements [23]. This is different from our current data, in which elevated CRP was a strong predictor of hypoxemia after hospitalization for COVID-19. These differences may be explained by including larger number of patients with more severe COVID-19 presentation and more pronounced inflammation at the time of admission to the hospital. Our findings highlight the implication of inflammation in disease progression and severity associated with SARS-CoV-2 infection in consistence with previous studies [24,25]. The induced inflammatory responses and cytokine release ultimately cause alveolar inflammation and reduced surfactant production, thereby accelerating pneumonic consolidation and subsequent hypoxemia [24,25]. Furthermore, low oxygen saturation may further exacerbate inflammation, thus creating an amplification circuit for worsening hypoxemia [16,17]. Our study is consistent with previous studies showing that increased inflammation is a hallmark of COVID-19 severity [1,15]. This is supported by randomized clinical trials demonstrating improved clinical outcomes with steroid therapy among hospitalized patients with COVID-19 and hypoxemia [26]. 

Additional important finding of our study is the role of body habitus, represented by BMI or BSA, in prediction of COVID-19-related hypoxemia. Morbid obesity has been shown to be an important risk factor for respiratory infections and for severe COVID-19 [27,28]. According to recent studies, morbid obesity, represented in our study by increased BMI, has been implicated in worse outcomes, including mortality, among patients diagnosed with COVID-19 [29,30,31]. This observation can be explained, in part, by higher prevalence of other cardiometabolic diseases and comorbidities; higher expression of angiotensin-converting enzyme 2 (ACE-2), an essential receptor for host cell infection with SARS-CoV-2; increased secretion of proinflammatory cytokine release from adipose tissues; and several adverse respiratory mechanical factors in obesity [29,32,33]. However, all previous studies have included only BMI as a surrogate of body size for assessing adverse outcomes associated with COVID-19. In the current study, we report, for the first time, on a significant correlation between BSA and low oxygen saturation, demonstrating that smaller body size is associated with higher oxygen saturation and that BSA has noninferior performance in the diagnosis of hypoxemia among patients admitted for COVID-19 as compared to BMI. Since BMI alone may not precisely assess the amount and distribution of adipose tissue compared to lean body mass, it may not reflect accurately relevant COVID-19 comorbidities, such as obesity, DM, and hypertension. Therefore, measurement of BSA, especially when combined with BMI, could be an additive risk marker in predicting risk of hypoxemia and cardiovascular complications seen in COVID-19 patients. Additionally, BSA may represent an accurate parameter of lung size and expansion; therefore, it may determine susceptibility to lung injury from COVID-19 infection in a more precise manner. 

In line with previous reports [16,34], the present data confirm that hypoxemia is associated with need for ventilatory support and development of ARDS, leading to increased in-hospital mortality. We further demonstrate that the presence of a larger number of risk factors identified for hypoxemia has an incremental prognostic value in terms of developing these respiratory complications. 

The current study confirmed that increased age, specifically 60 years old or older, was independently associated with in-hospital mortality in patients with COVID-19, similarly to previous reports [4,35,36]. Our analysis also demonstrated that pre-existing comorbidities, including chronic renal failure and HF, are associated with increased mortality. The association of these comorbidities with severe COVID-19 and mortality has also been replicated across major studies [1,2,3,4,5]. Our study showed a weak association of other comorbid conditions, such as DM and hypertension, which are recognized risk factors for severe infection, with mortality when analyzed individually. However, the risk of mortality was incremental in the presence of larger number of comorbidities, and patients with any of these comorbidities were found to be at markedly increased mortality risk as compared to those without any comorbidity. In addition, we found a trend for increased mortality in patients with obesity, but results were underpowered to achieve statistical significance. Although increased BMI was identified as an independent predictor of hypoxemia in hospitalized COVID-19 patients, thereby contributing to the increased need for ventilation in this cohort, BMI was not found to be a predictor of in-hospital mortality. These findings can be explained by the fact that obese individuals usually had multiple comorbidities, which could increase the risk of mortality on their own and confound the results. Moreover, as hypoxemia is a modifiable condition, risk factors of this intermediate outcome, such as high BMI or BSA, may not necessarily predict mortality when hypoxemia is managed early and aggressively to mitigate disease progression and death. 

The results of this analysis are subject to limitations inherent to the observational, retrospective, nonrandomized design of our study. Therefore, as in any observational study, we cannot rule out residual confounding and survival treatment selection. For example, as it was the “second wave” of the pandemic, no information was available on anosmia, hyposmia, or treatments for COVID-19 during hospitalization, which varied throughout the study period. As multiple testing was performed using many variables in our models, we cannot rule out an inflated rate of false-positive conclusions. However, in the regression models, we included the most clinically relevant variables that could be implicated in the development of hypoxemia. Although our data is based on ethnically diverse population living in Jerusalem (Arabs and Ashkenazi, Sephardic and Ethiopian Jews), it needs to be validated in other communities, such as in Asia and South America, where COVID-19 is still an expanding pandemic with limited resources. 

In conclusion, our double-center cohort analysis demonstrates that among patients hospitalized with moderate to severe COVID-19, older age, lymphopenia, and elevated CRP were independent predictors of hypoxemia and respiratory-related adverse outcomes. Both large body size measurements, as reflected by increased BMI or BSA, were associated with low oxygen saturation. Severe hypoxemia requiring ventilatory support and underlying comorbidities, including older age, chronic renal failure, and underlying HF, were identified as significant predictors of in-hospital mortality. In the context of the current, still-expanding COVID-19 pandemic in many developing countries, which struggle with scarce health and economic resources, detailed assessment of these risk factors at the time of hospitalization may be helpful in appropriate patient triage and early management of hypoxemia and may ultimately contribute to reducing the disease burden in these communities.

## Figures and Tables

**Figure 1 jcm-10-03581-f001:**
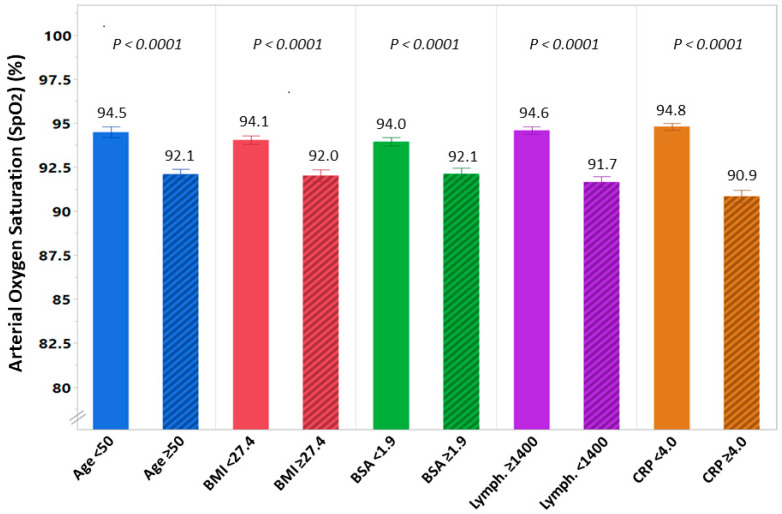
Differences in arterial oxygen saturation (SpO_2_) among patients hospitalized with COVID-19 stratified according to the independent predictors of low SpO_2_: age ≥ 50 vs. age < 50 years, high body mass index (BMI) (≥27.4 kg/m^2^) vs. low BMI (<27.4 kg/m^2^), high body surface area (BSA) (≥1.9 m^2^) vs. low BSA (<1.9 m^2^), low lymphocyte count (<1400 cells/μL) vs. high lymphocyte count (≥1400 cells/μL), and high c-reactive protein (CRP) (≥4.0 mg/dL) vs. low CRP (<4.0 mg/dL). SpO_2_ is presented in percentages.

**Figure 2 jcm-10-03581-f002:**
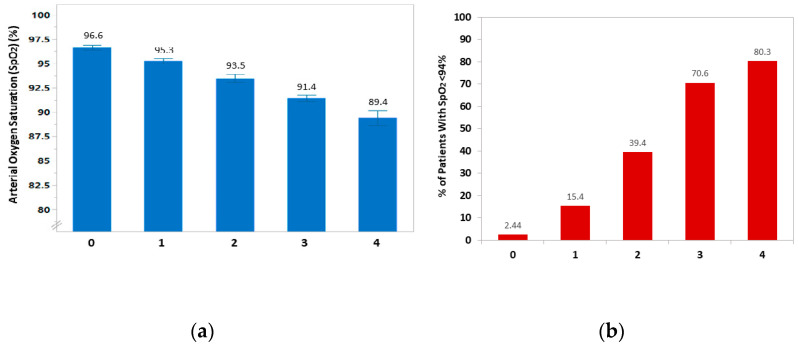
(**a**) Arterial oxygen saturation (SpO_2_) according to the number of risk factors (age ≥ 50 years, BMI ≥ 27.4 kg/m^2^, lymphocyte count <1400 cells/μL, and CRP ≥ 4.0 mg/dL) in each patient at the time of hospitalization with COVID-19. (**b**) The proportion of patients with SpO_2_ < 94% according to the number of these risk factors.

**Figure 3 jcm-10-03581-f003:**
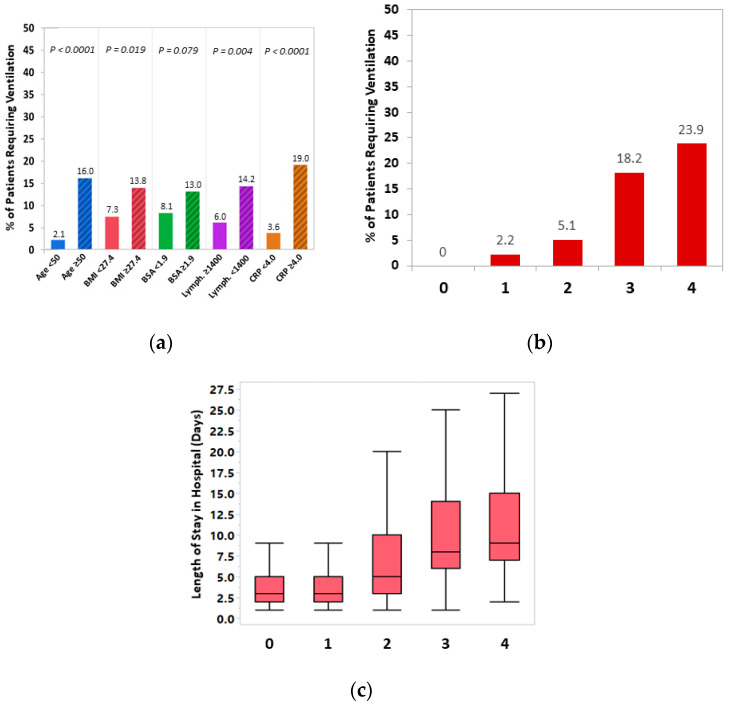
(**a**) Differences in the proportion of patients requiring in-hospital ventilation according to the risk factors identified for low SpO_2_ (age ≥ 50 years, BMI ≥ 27.4 kg/m^2^, lymphocyte count < 1400 cells/µL, and CRP ≥ 4 mg/dL). (**b**) Differences in the proportion of patients requiring in-hospital ventilation; and (**c**) differences in the length of stay (LOS) in hospital (presented in days) according to the number of these risk factors.

**Figure 4 jcm-10-03581-f004:**
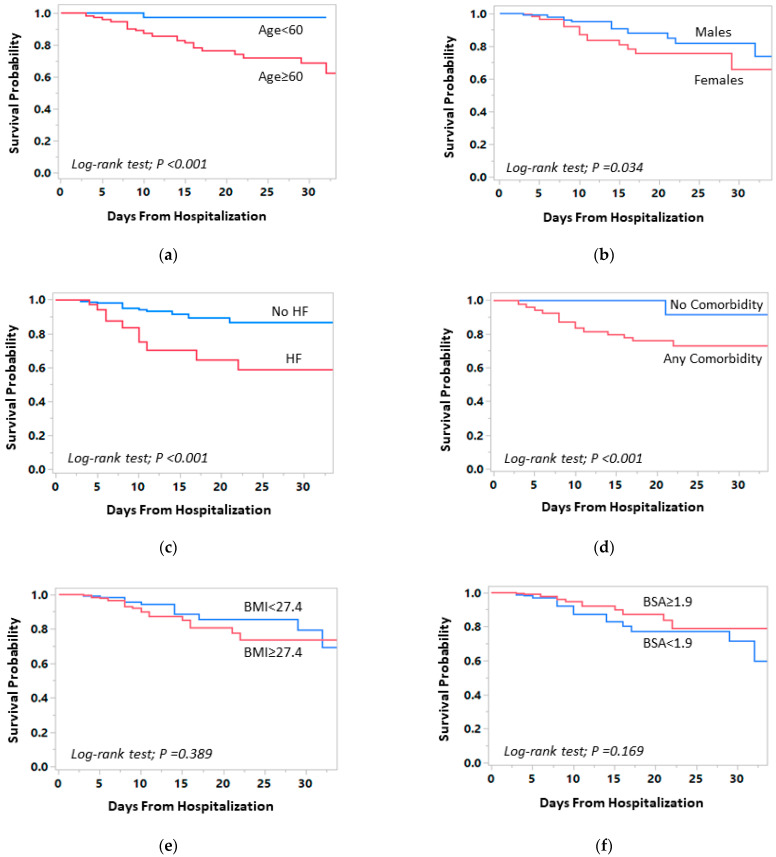
Kaplan–Meier curves for in-hospital mortality by demographic and clinical data comparing (**a**) adult patients 60 years or older and patients younger than 60 years, (**b**) females and males, (**c**) patients without and with underlying heart failure (HF), (**d**) patients without and with any comorbidity (defines as having ≥1 of the following: hypertension, diabetes mellitus, established cardiovascular disease, underlying HF, or chronic renal failure stage ≥4), (**e**) patients without and with high body mass index (BMI) (≥27.4 kg/m^2^), and (**f**) patients without and with high body surface area (BSA) (≥27.4 kg/m^2^).

**Table 1 jcm-10-03581-t001:** Demographic, clinical, and laboratory characteristics of patients hospitalized with COVID-19 comparing patients with and without low oxygen saturation.

Characteristic	All Patients	SpO_2_ ≥ 94%	SpO_2_ < 94%	*p*-Value
	*n* = 492	*n* = 271	*n* = 221	
Demographics				
Age, years, mean ± SD	55.9 ± 20.6	50.7 ± 21.2	62.3 ± 17.8	<0.001
Gender (male), *n* (%)	259 (52.6%)	124 (45.8%)	135 (61.1%)	<0.001
Height (m), mean ± SD	1.68 ± 0.09	1.68 ± 0.10	1.68 ± 0.08	0.919
Weight (kg), mean ± SD	79.2 ± 18.5	76.0 ± 18.6	83.1 ± 17.7	<0.001
BMI (kg/m^2^), mean ± SD	28.20 ± 6.10	27.13 ± 6.10	29.61 ± 5.92	<0.001
BSA (m^2^), mean ± SD	1.91 ± 0.24	1.87 ± 0.25	1.96 ± 0.23	<0.001
Current smoking, *n* (%)	33 (9.5%)	17 (9.2%)	16 (9.9%)	0.840
Ethnicity: Jews Arabs	431 (87.6%) 61 (12.4%)	237 (87.5%) 34 (12.6%)	194 (87.8%) 27 (12.2%)	0.912
Comorbidities				
Hypertension, *n* (%)	165 (42%)	68 (33.0%)	97 (51.1%)	<0.001
Diabetes mellitus, *n* (%)	113 (29%)	44 (22.2%)	69 (36.3%)	0.002
Hyperlipidemia, *n* (%)	105 (28%)	48 (24.0%)	57 (32.0%)	0.082
Cognitive decline, *n* (%)	68 (18.8%)	29 (15.2%)	39 (18.8%)	0.064
Atherosclerotic CVD, *n* (%)	64 (17.8%)	33 (17.0%)	31 (18.7%)	0.681
Lung dysfunction *, *n* (%)	43 (12.1%)	25 (12.4%)	18 (11.0%)	0.543
Heart failure, *n* (%)	44 (12.3%)	19 (10.2%)	25 (14.9%)	0.178
Atrial fibrillation/flutter, *n* (%)	34 (9.7%)	16 (8.7%)	18 (10.8%)	0.498
Malignancy, *n* (%)	17 (4.9%)	9 (5.0%)	8 (4.9%)	0.998
VTE, *n* (%)	14 (4.1%)	5 (2.7%)	9 (5.6%)	0.181
Immunosuppression, *n* (%)	14 (4.1%)	10 (5.5%)	4 (2.5%)	0.166
Autoimmune disease, *n* (%)	11 (3.2%)	7 (3.8%)	4 (2.5%)	0.493
Pulmonary hypertension, *n* (%)	8 (2.3%)	5 (2.7%)	3 (1.9%)	0.598
Renal replacement therapy, *n* (%)	5 (1.0%)	4 (1.5%)	1 (0.45%)	0.256
Medications				
Statins, *n* (%)	93 (25.6%)	37 (19.7%)	56 (32.0%)	0.007
ACE/ARB, *n* (%)	82 (22.8%)	34 (18.1%)	48 (27.9%)	0.027
Aspirin, *n* (%)	73 (20.1%)	29 (15.1%)	44 (25.7%)	0.012
Antiplatelet therapy (any), *n* (%)	95 (19.5%)	41 (15.4%)	54 (24.6%)	0.011
Beta blockers, *n* (%)	70 (19.1%)	32 (16.5%)	38 (22.1)	0.174
Insulin, *n* (%)	27 (7.7%)	4 (2.2%)	23 (13.7%)	<0.001
THRT, *n* (%)	48 (13.7%)	19 (10.3%)	29 (17.5%)	0.050
CCB, *n* (%)	47 (13.2%)	21 (11.2%)	26 (15.5%)	0.239
Diuretics, *n* (%)	47 (13.2%)	15 (8.1%)	32 (18.8%)	0.003
Bronchodilator, *n* (%)	23 (6.6%)	15 (8.2%)	8 (4.9%)	0.231
Anticoagulation, *n* (%)	11 (3.2%)	6 (3.3%)	5 (3.1%)	0.935
MRA, *n* (%)	9 (2.6%)	4 (2.2%)	5 (3.1%)	0.588
Laboratory				
Leukocytes, 10^3^/µL	6.4 (5.0–8.3)	6.4 (5.1–8.4)	6.6 (4.9–8.0)	0.642
Neutrophils, 10^3^/µL	4.4 (3.1–6.1)	4.0 (2.8–6.2)	4.7 (3.4–6.1)	0.036
Lymphocytes, 10^3^/µL	1.3 (0.9–1.8)	1.5 (1.1–2.2)	1.1 (0.8–1.4)	<0.001
Platelets, 10^3^/µL	193 (156–249)	195 (163–254)	189 (152–246)	0.324
Hemoglobin, g/dL, mean ± SD	13.5 ± 1.8	13.4 ± 1.9	13.6 ± 1.8	0.161
BUN, mg/dL	13 (10–20)	13 (9–17)	15 (10–23)	<0.001
Creatinine. mg/dL, median (IQR)	0.80 (0.63–0.98)	0.76 (0.61–0.95)	0.84 (0.65–1.11)	0.003
eGFR, mL/min, median (IQR)	105.9 (75.0–143.8)	114.2 (79.8–149.7)	100.6 (68.8–129.3)	0.012
Uric acid, mg/dL, mean ± SD	5.4 ± 2.4	5.8 ± 2.2	5.0 ± 2.7	0.090
Sodium, mmol/L, mean ± SD	137.0 ± 4.1	138.1 ± 3.1	135.4 ± 4.4	<0.001
Potassium, mmol/L, mean ± SD	4.0 ± 0.5	4.0 ± 0.5	4.0 ± 0.5	0.638
Phosphate, mg/dL, mean ± SD	3.1 ± 0.8	3.2 ± 0.9	3.0 ± 0.8	0.066
CRP, mg/dL, median (IQR)	4.2 (0.9–9.9)	1.5 (0.4–5.2)	7.3 (4.0–13.5)	<0.001
D-dimer, ng/mL, median (IQR)	719 (408–1188)	564 (306–1011)	896 (540–1416)	<0.001
Fibrinogen, mg/dL, mean ± SD	604.3 ± 173.5	534.8 ± 151.6	669.6 ± 167.5	<0.001
AST, IU/L, median (IQR)	32.0 (24.0–49.0)	27.0 (22.0–36.0)	39.5 (30.0–55.0)	<0.001
ALT, IU/L, median (IQR)	21.0 (14.0–34.0)	20.0 (14.0–34.0)	24.0 (15.0–36.5)	0.118
GGT, IU/L, median (IQR)	30.0 (18.0–56.0)	26.0 (16.0–44.0)	36.0 (21.5–71.0)	<0.001

Abbreviations: SD indicates standard deviation; IQR, interquartile range; SpO_2_, arterial oxygen saturation; BMI, body mass index; BSA, body surface area; CVD, cardiovascular disease; VTE, venous thromboembolism; ACE, angiotensin converting enzyme; ARB, angiotensin II receptor blocker; OHG, oral anti-hyperglycemic agent; THRT, thyroid hormone replacement therapy; CCB, calcium channel blocker; MRA, mineralocorticoid receptor antagonists; CIED, cardiac implantable electronic device; eGFR, estimated glomerular filtration rate; GGT, gamma-glutamyl transferase; AST, aspartate transaminase; ALT, alanine transaminase; CRP, c-reactive protein; BUN, blood urea nitrogen. * Lung dysfunction was defined as having a history of obstructive or/and restrictive lung disease.

**Table 2 jcm-10-03581-t002:** Univariable and Multivariable models of risk factors associated with low SpO_2_.

Characteristic	Univariable Model			Multivariable Model		
	Odds Ratio	95% CI	*p*-Value	Odds Ratio	95% CI	*p*-Value
Age (per year)	1.03	1.01 to 1.04	<0.001	1.02	1.01 to 1.03	<0.001
Gender (male)	1.86	1.30 to 2.67	<0.001	1.50	0.96 to 2.32	0.074
BMI (per 1 kg/m^2^)	1.08	1.04 to 1.011	<0.001	1.05	1.01 to 1.09	0.011
BSA (per 0.1 m^2^)	1.17	1.08 to 1.26	<0.001	1.16	1.05 to 1.27	0.003
Diabetes Mellitus	2.00	1.30 to 2.67	0.002			
Hypertension	2.11	1.41 to 3.18	<0.001			
Hyperlipidemia	1.49	0.95 to 2.34	0.083			
Cognitive decline	1.65	0.97 to 2.81	0.065			
Baseline creatinine (per 1 mg/dL)	1.37	0.98 to 1.93	0.068			
BUN (per 1 mg/dL)	1.03	1.01 to 1.05	0.001			
Lymphocyte count (per 10^3^/µL)	0.39	0.29 to 0.53	<0.001	0.58	0.41 to 0.81	0.002
CRP (per mg/dL)	1.15	1.11 to 1.20	<0.001	1.11	1.07 to 1.16	<0.001
D-dimer (per 0.1 ug/mL)	1.04	1.01 to 1.06	0.003			
Fibrinogen (per 10 ng/mL)	1.05	1.04 to 1.07	<0.001			
AST (per 10 IU/L)	1.12	1.04 to 1.21	0.004			
GGT (per 10 IU/L)	1.04	1.01 to 1.07	0.017			
ACE-I/ARB therapy	1.75	1.06 to 2.89	0.027			
THRT	1.85	0.99 to 3.44	0.052			
Diuretic therapy	2.63	1.37 to 5.05	0.004			

Abbreviations: BSA indicates body surface area; BMI, body mass index; CI, confidence interval; GGT, gamma-glutamyl transferase; AST, aspartate transaminase; CRP, c-reactive protein; BUN, blood urea nitrogen; ACE, angiotensin converting enzyme; ARB, angiotensin II receptor blocker; THRT, thyroid hormone replacement therapy. Multivariable model (with BSA included): R^2^ = 0.193, *p* < 0.001. Multivariable model (with BMI included): R^2^ = 0.189, *p* < 0.001.

**Table 3 jcm-10-03581-t003:** Univariate and multivariate Cox regression analysis of risk factors of in-hospital mortality among patients hospitalized with COVID-19.

Risk Factor	Univariate Analysis		Multivariate Analysis *	
	HR (95% CI)	*p* Value	HR (95% CI)	*p* Value
Age ≥60 years	15.7 (2.1 to 115.8)	0.007		
Sex (females vs. males)	2.1 (1.1 to 4.3)	0.039		
Hypertension	2.0 (0.9 to 4.6)	0.090	1.0 (0.5 to 2.4)	0.950
Diabetes mellitus	1.3 (0.6 to 2.8)	0.493	1.0 (0.5 to 2.2)	0.916
Established CVD	2.3 (1.1 to 5.1)	0.036	1.7 (0.8 to 3.8)	0.187
Chronic heart failure	4.1 (1.8 to 9.1)	<0.001	2.7 (1.2 to 6.0)	0.016
Chronic renal failure ^#^	4.3 (2.0 to 9.3)	<0.001	2.8 (1.3 to 6.0)	0.012
Any Comorbidity ^$^	17.0 (2.3 to 126.7)	0.006	9.4 (1.2 to 75.4)	0.035
BMI ≥27.4 kg/m^2^	1.4 (1.7 to 2.8)	0.392	1.4 (0.7 to 2.9)	0.337
BSA ≥ 1.9 m^2^	0.62 (0.31 to 1.20)	0.175	0.86 (0.42 to 1.80)	0.682
CRP ≥4 mg/dL	1.0 (0.5 to 2.2)	0.954	1.1 (0.5 to 2.4)	0.794
Hypoxemia requiring ventilatory support	2.2 (1.0 to 4.6)	0.047	2.3 (1.1 to 4.9)	0.036

Abbreviations: BMI indicates body mass index; BSA, body surface area; CI, confidence interval; CRP, c-reactive protein; CVD, cardiovascular disease; HR, hazard ratio. * Adjusted for age and sex. ^#^ Chronic renal failure stage 4 or 5 (i.e., eGFR < 30 mL/min). ^$^ Presence of ≥1 comorbidity (hypertension, diabetes mellitus, established cardiovascular disease, heart failure, or chronic renal failure stage ≥4).

## Data Availability

The data presented in this study are available on request from the corresponding author. The data are not publicly available due to patient privacy.

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
