# Peer review of "Predictors of Hypoxemia and Related Adverse Outcomes in Patients Hospitalized with COVID-19: A Double-Center Retrospective Study"

_jcm, 2021, doi:10.3390/jcm10163581_

Round 1

Reviewer 1 Report

Dear authors,

congratulations to you nice work which will contribute to our growing knowlwdge in the filed pf coVid.

There are a couple of questions which are of interest:

  1. Why do you choose 94% as a cut off value instead of 90% or 95% (Ref. 7/8), is it helpful to bring in a second threshold like 90% as suggested by some other groups?
  2. I think the results section needs some re-structuring in terms of making it better readable for the readers. There is a lot of indormation which needs more priorization

Author Response

Responses to Reviewers

We appreciate the constructive comments of the Reviewers and have modified the paper accordingly. Below are the point-by-point responses.

Reviewer #1:

Dear authors,

Congratulations to your nice work which will contribute to our growing knowledge in the field of COVID.

RE: We thank the Reviewer for the positive comments.

There are a couple of questions which are of interest:

1. Why do you choose 94% as a cut off value instead of 90% or 95% (Ref. 7/8), is it helpful to bring in a second threshold like 90% as suggested by some other groups?

RE: We appreciate the Reviewer`s comments. The cutoff value of 94% has been used to define severe COVID-19 infection in several studies. We have cited references #8 and #9 in the manuscript, but other studies (Gandhi RT, et al. Mild or moderate Covid-19; NEJM 2020 Oct 29;383(18):1757-1766; and Wu Z, et al. Characteristics of and important lessons from the coronavirus disease 2019 (COVID-19) outbreak in China: summary of 72,314 cases from the Chinese Center for Disease Control and Prevention. JAMA 2020;323;1239-1242) as well as the GS-US-540-5773 trial for assessing the benefit of remdesivir in patients with severe COVID-19 (Goldman GD et al. Remdesivir for 5 or 10 days in patients with severe Covid-19. NEJM 2020 Nov 5;383(19):1827-1837) have used this oxygen saturation cutoff value to determine disease severity. Although different oxygen saturation cutoffs have been described previously, we elected to use the 94% value to seek predictors of early hypoxemia as an intermediate outcome that might be associated with subsequent adverse clinical outcomes in consistence with these studies.

2. I think the results section needs some re-structuring in terms of making it better readable for the readers. There is a lot of information which needs more prioritization

RE: We thank the Reviewer for this excellent comment with which we agree. We have shortened the results section considerably and revised it to present the main study findings. In the revised manuscript, we have also moved the section on BSA and BMI to the supplementary materials to keep the results section more concise and clearer for the readership.

Reviewer 2 Report

Review report for the manuscript “Predictors of Hypoxemia and Related Adverse Outcomes in Patients Hospitalized With COVID-19: A Double-Center Retrospective Study”,  submitted by Asleh B et al.

I thank the journal to allow me to review this manuscript. Furthermore, I congratulate the authors for conducting this research study.

Summary

Asleh et al. conducted a retrospective cohort study in 492 hospitalized patients with COVID-19. The purpose of the study was to assess risk factors associated with hypoxemia (SpO2<94%), which has been suggested to be useful as a surrogate endpoint for the worse progression of COVID-19. The authors identified by multiple regression analysis and ROC analyses, 4 risk factors to develop hypoxemia: age ≥50 years; BMI ≥27.4 kg/m2; lymphocyte count <1400 cells/ul; and CRP ≥4 mg/dl. These risk factors combined together, resulted in an accuracy to predict hypoxemia of 0.81 (no 95%CI presented). The identified risk factors might be of value to improve the triage of patients and to earlier start supplemental oxygen therapy in those, which are at higher risk for hypoxemia and worse COVID-19 progression.

The manuscript is well written and of some clinical importance, however, the extensive length of the results are demanding and require some rephrasing, furthermore, predefined and exploratory analysis are not well defined. Other major and minor concerns are outlined below.

Major concerns

  1. The authors suggest to use the risk factors to better triage the patients based on the risk for developing hypoxemia and subsequent worse progression of COVID. However, it is unclear when hypoxemia was identified. In the text, line 274 the authors write “at the time of admission with COVID-19”. However, when hypoxemia is at the time of admission, then it seems not be necessary to triage for risk factors of hypoxemia, then we can directly triage by hypoxemia. Therefore, the logical time line of the risk factors and developing hypoxemia is unclear. It would be interesting to have information of the duration until the patients developed hypoxemia. Maybe this would strengthen the fact that the identified factors are actually risk factors for developing hypoxemia, followed by worse clinical outcomes. Then a triage would be of value.
  2. The manuscript does not provide enough information on how exactly the found risk factors could change clinical practice. It would be of value when the authors would describe, how the triage would work in reality. They write at line 342 “…and aggressive treatment with oxygen supplementation to mitigate disease progression”. What does that mean? Do the authors suggest to give supplemental oxygen to normoxic patients based on the risk assessment score? Please elaborate this statement and how the application of the risk factors would work in reality.
  3. The identified risk factors – age, BMI, Lymphocyte count and CRP are not further discussed why they are risk factors for hypoxemia. Are they plausible? Have they been identified in other diseases/conditions as risk factors? Please discuss them in the context of the literature.
  4. The manuscript and results are extensive and the reviewer feels that some parts are not focusing on the research question. The manuscript and conclusion could be improved by streamlining the manuscript by focusing on hypoxemia and the pre-defined research question. The extensive discussion and analysis around BSA and BMI were not a priori of interest, therefore, this could be reduced (section 3.2 or parts of the discussion and figures in the supplement).
  5. Furthermore, the tables contain many variables and therefore, multiple testing might be a problem in this manuscript. It would be good to have statistics controlled for multiple testing, or that the authors add this limitation to the discussion section.

Minor concerns

Abstract

  1. The abstract lacks numbers and statistics. Please change the abstract from qualitative to quantitative, as done in the results of the manuscript.
  2. Explain the abbreviation BSA on first mentioning.
  3. Focus in the abstract on risk factors for hypoxemia and COVID. Please delete line 28 to 29 regarding BMI and BSA.

Introduction

  1. Line 41, delete double space
  2. Line 58, typo
  3. Line 69 to 72, please rephrase the sentence, it is difficult to read. What was the purpose of the study?

Methods

  1. Line 79, typo
  2. Line 80 add ethic committee number
  3. Line 82, delete double space
  4. Line 82 to 83, please relocate to statistics.
  5. Line 85, typo

Results

  1. Please do not replicate numbers presented in the tables. The results are extensive and redundant, therefore, delete all numbers. This will reduce the text and improve the reading. Numbers and information from demographics can remain in the text; this is helpful to not switch to the table for demographic information.
  2. Line 136, define DM
  3. Line 158, define BUN
  4. Table 1: What is lung dysfunction? Define it in the legend, or delete it.
  5. Table 1: Smoking is not a disease. Is it current smoking? Please add this information to the demographics.
  6. Table 1: There are different formatting styles in the table. Please unify.
  7. Table 1: The table is extensive and not all information is of value. Please consider to reduce it. For example, delete “Previous interventions”
  8. Section 3.2 seems to me out of the scope of the manuscript. This exploratory analysis should be reduced and should be relocated after the regression analysis results. It is unclear, why you did a subgroup analysis based on median, when you will stratify them by ROC analysis, as initially proposed.
  9. The reviewer feels that the complete discussion with BSA and BMI seems less relevant. Unless the authors can argument that BSA is a better predictor for hypoxemia, then they should emphasize this and go along with BSA. Otherwise, BMI is easier to measure and can be preferred, no need to discuss BSA.
  10. Section 3.3 is a bit misleading, since these parameters are not all significantly associated with hypoxemia (P values above 0.05). These parameters are just the ones with a P<0.1 in the univariate regression analysis. Therefore, please rephrase this section.
  11. Table 2: Write 95% CI with a “to” between numbers: 1.01 to 1.03. This improves readability.
  12. Table 2: It would be of value to keep gender in the multiple regression model. This is an important predictor, unless the authors can show that R2 is lower with gender inside the model.
  13. Line 229: What means “lieu”? Typo?
  14. Line 243 to 245: please provide the 95% CI of the AUC numbers 0.81 and 0.80, respectively.
  15. Line 255: delete double space
  16. Figure 3: The three panels are differently formatted. Please unify font size, scale of the window ect.
  17. Line 314 and below covering the Kaplan Meier curves. It is surprising that you use other predictors for the survival analysis than identified by the Regression and ROC analysis. It would be interesting to see if the risk factors do also predict survival of COVID patients (not only hypoxemia). Otherwise, this section is off topic and is not in line with the research question of the manuscript. I would propose to perform the survival analysis with the obtained cut-off values of the risk factors. Additionally, also add the KM for hypoxemia, as a reference.

Discussion

  1. Line 376 delete comma
  2. Line 375 You describe different findings between your study and a study in Japanese. Why was there a difference, can you speculate?
  3. Line 378 to 379, how did you assess that CRP is the strongest predictor?
  4. Line 381, double space
  5. Line 407, “BSA and BMI might be additive in predicting risk of hypoxemia” You have not shown an additive effect. In the manuscript you have used them interchangeable. Please rephrase.
  6. Line 418 double space
  7. Limitations: Add the limitation of multiple testing.
  8. Line 444, delete “in” hospitalized…
  9. Please rephrase the conclusion by starting what you have found based on the initial research question. Then please continue with the findings of the exploratory analysis and what these results mean in the clinical practice.

Author Response

Responses to Reviewers

We appreciate the constructive comments of the Reviewers and have modified the paper accordingly. Below are the point-by-point responses.

Reviewer #2:

I thank the journal to allow me to review this manuscript. Furthermore, I congratulate the authors for conducting this research study.

 Summary

Asleh et al. conducted a retrospective cohort study in 492 hospitalized patients with COVID-19. The purpose of the study was to assess risk factors associated with hypoxemia (SpO2<94%), which has been suggested to be useful as a surrogate endpoint for the worse progression of COVID-19. The authors identified by multiple regression analysis and ROC analyses, 4 risk factors to develop hypoxemia: age ≥50 years; BMI ≥27.4 kg/m2; lymphocyte count <1400 cells/ul; and CRP ≥4 mg/dl. These risk factors combined together, resulted in an accuracy to predict hypoxemia of 0.81 (no 95%CI presented). The identified risk factors might be of value to improve the triage of patients and to earlier start supplemental oxygen therapy in those, which are at higher risk for hypoxemia and worse COVID-19 progression.

The manuscript is well written and of some clinical importance, however, the extensive length of the results are demanding and require some rephrasing, furthermore, predefined and exploratory analysis are not well defined. Other major and minor concerns are outlined below.

RE: We thank the Reviewer for the positive comments.

Major concerns

  1. The authors suggest to use the risk factors to better triage the patients based on the risk for developing hypoxemia and subsequent worse progression of COVID. However, it is unclear when hypoxemia was identified. In the text, line 274 the authors write “at the time of admission with COVID-19”. However, when hypoxemia is at the time of admission, then it seems not be necessary to triage for risk factors of hypoxemia, then we can directly triage by hypoxemia. Therefore, the logical time line of the risk factors and developing hypoxemia is unclear. It would be interesting to have information of the duration until the patients developed hypoxemia. Maybe this would strengthen the fact that the identified factors are actually risk factors for developing hypoxemia, followed by worse clinical outcomes. Then a triage would be of value.

 RE: We appreciate the Reviewer`s comment and apologize for not clarifying this in the manuscript. We have determined hypoxemia as developing blood oxygen saturation <94% within the first 48 hours of arriving to the emergency department with COVID-19. We have clarified this in the revised manuscript. Indeed, the use of the independent risk factors at arrival to the hospital is aimed at predicting early hypoxemia and thus it is of clinical value because early triage of patients with risk factors for hypoxemia, close monitoring, and early oxygen supplementation may reduce the risk of severe hypoxemia and subsequent adverse clinical events in this group as pointed out in the manuscript.

  1. The manuscript does not provide enough information on how exactly the found risk factors could change clinical practice. It would be of value when the authors would describe, how the triage would work in reality. They write at line 342 “…and aggressive treatment with oxygen supplementation to mitigate disease progression”. What does that mean? Do the authors suggest to give supplemental oxygen to normoxic patients based on the risk assessment score? Please elaborate this statement and how the application of the risk factors would work in reality.

RE: We thank the Reviewer for this valuable comment. Identifying risk factors for hypoxemia is important for triaging these patients to the intensive care unit with close follow-up and monitoring for possible deterioration. These patients are at higher risk for developing hypoxemia and ventilatory support. Patients with low risk for hypoxemia based on these risk factors are at lower risk of deterioration during hospitalization and may be discharged early or at least do not need intensive care unit, particularly in hospitals with limited ICU beds. Initiation of oxygen supplementation in normoxic patients is not recommended and cannot be supported by the currently available data. However, studies have shown that hypoxia begets further hypoxia in COVID-19 patients (Somers VK et al. Progressive Hypoxia. Mayo Clinic Proceedings 2020, 95, 2339–2342, and Eltzschig HK et al. Hypoxia and Inflammation. New England Journal of Medicine 2011), and thus early imitation of oxygen supplementation in patients with mild hypoxemia may attenuate disease progression and need for mechanical ventilatory support in the ICU setting. This has been elaborated in the revised manuscript for emphasizing the clinical value of the current study.

  1. The identified risk factors – age, BMI, Lymphocyte count and CRP are not further discussed why they are risk factors for hypoxemia. Are they plausible? Have they been identified in other diseases/conditions as risk factors? Please discuss them in the context of the literature.

RE: We appreciate the Reviewer`s comment. In the revised manuscript, we have included potential explanations and mechanisms underlying the association of these risk factors, including inflammation (represented by elevated CRP and lymphopenia), obesity and advanced age, with worse clinical course in COVID-19. In the discussion section we have added the following: “The induced inflammatory responses and cytokine release ultimately cause alveolar inflammation and reduced surfactant production, thereby accelerating pneumonic consolidation and subsequent hypoxemia. Furthermore, low oxygen saturation may further exacerbate inflammation, thus creating an amplification circuit for worsening hypoxemia. Our study is consistent with previous studies showing that increased inflammation is a hallmark of COVID-19 severity. This is supported by randomized clinical trials demonstrating improved clinical outcomes with steroid therapy among hospitalized patients with COVID-19 and hypoxemia.” For inflammation, and “This observation can be explained, in part, by higher prevalence of other cardiometabolic diseases and comorbidities, higher expression of angiotensin-converting enzyme 2 (ACE-2), an essential receptor for host cell infection with SARS-CoV-2, increased secretion of proinflammatory cytokine release from adipose tissues, and several adverse respiratory mechanical factors in obesity” for obesity.

  1. The manuscript and results are extensive and the reviewer feels that some parts are not focusing on the research question. The manuscript and conclusion could be improved by streamlining the manuscript by focusing on hypoxemia and the pre-defined research question. The extensive discussion and analysis around BSA and BMI were not a priori of interest, therefore, this could be reduced (section 3.2 or parts of the discussion and figures in the supplement).

RE: We thank the Reviewer for this excellent comment. We have shortened the results section considerably and revised it to present the main study findings. In the revised manuscript, we have also moved the section on BSA and BMI to the supplementary materials to keep the results section more concise and clearer for the readership. In the discussion section, we have elaborated on the underlying mechanisms that may explain the involvement of the identified risk factors in the development of COVID-19-related hypoxemia and clinical deterioration. Furthermore, as recommended by the reviewers, we have revised the conclusion section to better reflect the main study findings. BSA was non-inferior to BMI in predicting hypoxemia and we feel it is important to state it in the manuscript as additional reliable body habitus measure in hospitalized COVID-19 patients.

  1. Furthermore, the tables contain many variables and therefore, multiple testing might be a problem in this manuscript. It would be good to have statistics controlled for multiple testing, or that the authors add this limitation to the discussion section.

RE: We agree with this point that multiple testing may increase the probability of false-positive findings. However, in the regression models, we have included the most clinically important variables that may be implicated in the development of hypoxemia in the setting of COVID-19 infection. As suggested by the reviewer, we have added in the limitation section as follows: “As multiple testing was performed using many variables in our models, we cannot rule out an inflated rate of false-positive conclusions. However, in the regression models, we included the most clinically relevant variables that could be implicated in the development of hypoxemia.

Minor concerns

 Abstract

  1. The abstract lacks numbers and statistics. Please change the abstract from qualitative to quantitative, as done in the results of the manuscript.

RE: We have added numbers and statistics in the abstract as recommended.  

  1. Explain the abbreviation BSA on first mentioning.

RE: BSA was defined in the abstract and first mentioning in the text. 

  1. Focus in the abstract on risk factors for hypoxemia and COVID. Please delete line 28 to 29 regarding BMI and BSA.

RE: We have deleted these lines in the abstract as recommended. We have inserters numbers and statistics instead as recommended.   

Introduction 

  1. Line 41, delete double space

RE: Corrected.  

  1. Line 58, typo

 RE: Corrected

  1. Line 69 to 72, please rephrase the sentence, it is difficult to read. What was the purpose of the study?

RE: The sentence was clarified as suggested. In the revised manuscript we have rephrases it as follows: “Therefore, in the present double-center study, we sought to investigate whether there might be unique predictors of the development of hypoxemia during hospitalization due to COVID-19 and to examine their associations with subsequent adverse clinical outcomes.

 Methods

  1. Line 79, typo

RE: Corrected

  1. Line 80 add ethic committee number

RE: We thank the Reviewer for this excellent comment. Ethic numbers of the two medical centers have been added in the revised manuscript.

  1. Line 82, delete double space

RE: Corrected

  1. Line 82 to 83, please relocate to statistics.

RE: This sentence was moved to statistics section as recommended.

  1. Line 85, typo

RE: Corrected

Results

  1. Please do not replicate numbers presented in the tables. The results are extensive and redundant, therefore, delete all numbers. This will reduce the text and improve the reading. Numbers and information from demographics can remain in the text; this is helpful to not switch to the table for demographic information.

RE: We appreciate the Reviewer`s comment. We have deleted numbers from the text and referred the readers to Table 1. As suggested, we have kept the numbers and statistics for important demographic data.

  1. Line 136, define DM

RE: DM was first mentioned and defined in the introduction (line 34).

  1. Line 158, define BUN

RE: BUN (blood urea nitrogen) was defined as suggested.

  1. Table 1: What is lung dysfunction? Define it in the legend, or delete it.

RE: Lung dysfunction was defined as having an underlying diagnosis of chronic obstructive or/and restrictive lung disease. This has been added in the table footnotes. 

  1. Table 1: Smoking is not a disease. Is it current smoking? Please add this information to the demographics.

RE: It is current smoking. This has been clarified and moved to demographics as recommended by the Reviewer. 

  1. Table 1: There are different formatting styles in the table. Please unify.

RE: Corrected.

  1. Table 1: The table is extensive and not all information is of value. Please consider to reduce it. For example, delete “Previous interventions”

RE: We appreciate the Reviewer`s comment. We have deleted “Previous intervention” and other less important variables as suggested. 

  1. Section 3.2 seems to me out of the scope of the manuscript. This exploratory analysis should be reduced and should be relocated after the regression analysis results. It is unclear, why you did a subgroup analysis based on median, when you will stratify them by ROC analysis, as initially proposed.

RE: We appreciate the Reviewer`s comment. As recommended by the Reviewer, this section has been removed. Specific parts on the correlation analysis of BSA and BMI with SpO2 as a continuous variable have been moved to the supplementary materials as recommended.

  1. The reviewer feels that the complete discussion with BSA and BMI seems less relevant. Unless the authors can argument that BSA is a better predictor for hypoxemia, then they should emphasize this and go along with BSA. Otherwise, BMI is easier to measure and can be preferred, no need to discuss BSA.

RE: This is an important point. BSA has become a robust measure for indexing hemodynamic parameters. However, its prognostic value as compared to BMI in patients hospitalized with COVID-19 has not yet been studied, hence it was an additional aim of this study. BMI as a risk factor for mortality in COVID-19 patients has been previously examined and we sought to examine if BSA is also reliable and perhaps additive to BMI. Therefore, we further sought to examine whether BSA is positively correlated with low oxygen saturation levels in patients hospitalized with COVID-19 and whether BSA might be a better predictor of hypoxemia as compared to BMI. This was the rationale for including both variables and compare their performances in this setting. We found that BSA was as good as BMI in predicting hypoxemia. Although not superior to BMI, the finding that BSA is also an independent predictor of hypoxemia is of clinical value and additive to the literature, particularly with the more expanding clinical use of BSA in body size and hemodynamic indexing. Moreover, we have shown that oxygen saturation was the lowest when both BSA and BMI were elevated, while in approximately 20% of the patients where BMI and BSA were discordant, no significant decrease in oxygen saturation was observed as compared to patients who had both low BMI and low BSA (Suppl. Figure IV). However, we understand the Reviewer`s point and have shortened this section in the revised manuscript.

  1. Section 3.3 is a bit misleading, since these parameters are not all significantly associated with hypoxemia (P values above 0.05). These parameters are just the ones with a P<0.1 in the univariate regression analysis. Therefore, please rephrase this section.

RE: We appreciate the Reviewer`s comment with which we agree. The section has been revised to include only variables with significant associations as other variables with P values between 0.05 and 0.10 were not significantly associated with hypoxemia but they were included in the multivariable regression models.   

  1. Table 2: Write 95% CI with a “to” between numbers: 1.01 to 1.03. This improves readability.

RE: Corrected.

  1. Table 2: It would be of value to keep gender in the multiple regression model. This is an important predictor, unless the authors can show that R2 is lower with gender inside the model.

RE: We appreciate the Reviewer`s comment. In the multivariable regression model, the association  of gender with hypoxemia was attenuated (OR 1.50, 95% CI: 0.96 to 2.32; P=0.074). No remarkable change in R2 was observed in the presence or absence of gender in the regression model. We have updated Table 2 in the revised manuscript to include gender in the multivariable regression model and included this information in the results section as follows: “The association between gender and hypoxemia was attenuated after adjustment for other variables (OR 1.5, 95% CI: 0.96-2.32; P=0.074, for males vs. females).”    

  1. Line 229: What means “lieu”? Typo?

 RE: “in lieu of” means “instead of”. We have changed it to “instead” to make it clearer to the readership.

  1. Line 243 to 245: please provide the 95% CI of the AUC numbers 0.81 and 0.80, respectively.

RE: We thank the Reviewer for this excellent comment. As suggested, wee have included the 95% CI for these AUC numbers. In the revised manuscript we have added the following: “AUC 0.81 (95% CI: 0.77-0.85) and AUC 0.80 (95% CI: 0.76-0.84)”.

  1. Line 255: delete double space

RE: Corrected.

  1. Figure 3: The three panels are differently formatted. Please unify font size, scale of the window ect.

RE: We thank the reviewer for this comment. We have unified the font size and scales of the 3 panels and submitted the revised figure as suggested. We anticipate further processing to optimize the presentation of all figures in the manuscript by the publisher.      

  1. Line 314 and below covering the Kaplan Meier curves. It is surprising that you use other predictors for the survival analysis than identified by the Regression and ROC analysis. It would be interesting to see if the risk factors do also predict survival of COVID patients (not only hypoxemia). Otherwise, this section is off topic and is not in line with the research question of the manuscript. I would propose to perform the survival analysis with the obtained cut-off values of the risk factors. Additionally, also add the KM for hypoxemia, as a reference.

RE: We appreciate the Reviewer`s comments and respectfully disagree with them. We have shown that hypoxemia is an independent predictor of survival. The identified risk factors of hypoxemia (age, BMI/BSA, CRP, etc.) as well as hypoxemia have been analyzed for associations with mortality.  In the last paragraph of the Results, we included the following: “High BMI, BSA or CRP were not associated with in-hospital mortality. However, hypoxemia requiring ventilation was found to be significantly associated with increased risk of mortality (adjusted HR 2.3, 95% CI: 1.1-4.9; P =0.036) (Table 2).” Besides hypoxemia (which was the focus of the study), we have identified other significant mortality risk factors (comorbidities, etc.), and believe that they are important to include when performing Cox regression for mortality analysis in our cohort. In the discussion, we have provided potential explanations why hypoxemia requiring ventilatory support, but not BMI/BSA or CRP, were associated with increased in-hospital mortality. KM can be generated to assess differences in risks only when variables or interventions are included at baseline. Because hypoxemia is an intermediate outcome and not a baseline characteristic (developed at various time points after admission to the emergency department), KM curves for hypoxemia are inappropriate.  

 Discussion

34. Line 376 delete comma

RE: Corrected.

35. Line 375 You describe different findings between your study and a study in Japanese. Why was there a difference, can you speculate?

RE: We appreciate the Reviewer`s comment. We have elaborated on the differences between the 2 studies in the revised version as follows: “In 84 Japanese patients admitted with mild to moderate COVID-19, advanced age, lymphopenia, and obesity, but not CRP, were associated with increased oxygen requirement. This is different from our current data in which elevated CRP was a strong predictor of hypoxemia after hospitalization for COVID-19. These differences may be explained by including larger number of patients with more severe COVID-19 presentation and more pronounced inflammation at the time of admission to the hospital. Our findings highlight the implication of inflammation in disease progression and severity associated with SARS Cov-2 infection in consistence with previous studies.

36. Line 378 to 379, how did you assess that CRP is the strongest predictor?

RE: We appreciate the Reviewer`s comment. This was assessed based on the multivariable regression estimates. However, since the prediction impact depends on the magnitude of increase in each predictor (which is different among the various predictors), we have changed this statement to “a strong predictor” rather than “the strongest predictor.”   

37. Line 381, double space

RE: Corrected.

38. Line 407, “BSA and BMI might be additive in predicting risk of hypoxemia” You have not shown an additive effect. In the manuscript you have used them interchangeable. Please rephrase.

RE: We appreciate the Reviewer`s comment with which we agree. BSA was not superior to BMI in predicting hypoxemia and other adverse related outcomes. This is represented by similar AUC for predicting low oxygen saturation when BSA and BMI were included independently. This has been corrected in the revised manuscript. However, when combined, we have observed that patients with high BMI and low BSA (and vice versa) had more favorable oxygen saturation than those with high BMI and BSA (Suppl. Figure IV).

39. Line 418 double space

RE: Corrected.

40. Limitations: Add the limitation of multiple testing.

RE: The following has been added in the limitations section in the revised manuscript “As multiple testing was performed using many variables in our models, we cannot rule out an inflated rate of false-positive conclusions. However, in the regression models, we included the most clinically relevant variables that could be implicated in the development of hypoxemia.

41. Line 444, delete “in” hospitalized…

RE: Corrected.

42. Please rephrase the conclusion by starting what you have found based on the initial research question. Then please continue with the findings of the exploratory analysis and what these results mean in the clinical practice.

RE: WE appreciate the Reviewer`s comment. The conclusion has been rephrased in the revised manuscript as suggested.

Round 2

Reviewer 2 Report

Round 2: Review report for the manuscript “Predictors of Hypoxemia and Related Adverse Outcomes in Patients Hospitalized With COVID-19: A Double-Center Retrospective Study”,  submitted by Asleh B et al.

I thank the authors for their thoughtful revision, improvements and comments. The manuscript has improved in content and readability. I have some minor comments, which should be addressed before proceeding:

  1. Please add to the abstract that patients were categorised by SpO2 <94% within the first 48 hours at the emergency department.
  2. Table 3: Please add a “to” instead of a “-“ in the 95% confidence intervals.
  3. A detail, but please use the same number of decimals within a variable. For example, table 3 DM: 1.3 (0.61 to 2.8). Harmonize to either 1.30 (0.61 to 2.80) or to 1.3 (0.6 to 2.8). Please re-check all tables.

Author Response

I thank the authors for their thoughtful revision, improvements and comments. The manuscript has improved in content and readability.

RE: We than the Reviewer for these positive comments and appreciate his/her thorough and constructive review of our manuscript. 

I have some minor comments, which should be addressed before proceeding:

  1. Please add to the abstract that patients were categorised by SpO2 <94% within the first 48 hours at the emergency department.

RE: This information has been added to the abstract of the revised manuscript as suggested. In the revised manuscript we have added the following "Patients were categorized as those who developed reduced (<94%) versus preserved (≥94%) arterial oxygen saturation (SpO2) within the first 48 hours of arrival to the emergency department." 

  1. Table 3: Please add a “to” instead of a “-“ in the 95% confidence intervals.

RE: Corrected. 

  1. A detail, but please use the same number of decimals within a variable. For example, table 3 DM: 1.3 (0.61 to 2.8). Harmonize to either 1.30 (0.61 to 2.80) or to 1.3 (0.6 to 2.8). Please re-check all tables.

RE: We appreciate the Reviewer`s comment. As suggested, we have reviewed all numbers in the text and tables to ensure that the same number of decimals is used within each variable in the revised manuscript.